# Household food insecurity and its association with academic performance among primary school adolescents in Hargeisa City, Somaliland

Sagal Mohamed Adam[1]*, Melese Sinaga Teshoma[1], Awale Sh. Dahir Ahmed[2], Dessalegn Tamiru[1]

1 Department of Nutrition and Dietetics, Faculty of Public Health, Health Institute, Jimma University, Jimma, Ethiopia, 2 Department of Public Health, Faculty of Allied Health Science, Daffodil International University, Dhaka, Bangladesh

* sagalsama24@gmail.com

## Abstract

### Background

Academic achievement is crucial for the social and economic development of young people and determines the quality of education of a nation. According to different studies, food insecurity adversely affects children's health, nutrition, and subsequent decline in academic performance by impairing students' ability to learn and therefore affects the achievement of the Sustainable Development Goals (SDGs) of quality education and lifelong learning opportunities for all. To provide evidence on the association of food insecurity with academic performance is necessary. The current study assessed household food insecurity and its association with academic performance among primary school adolescents in Hargeisa City, Somaliland.

### Methods

A school-based cross-sectional study was conducted among 630 primary school adolescents from December 2021 to March 2022. Students were selected using a multistage sampling technique. An interviewer-administered questionnaire was used to collect data on household food security and socio-demographic variables and entered into Epi data version 3.1. The data was exported to SPSS version 26 for descriptive and multivariable logistic regression analysis. Odds ratios and their 95% confidence intervals together with p <0.05 were utilized to declare statistical significance.

### Results

The prevalence of food insecurity among school adolescents was 59.21%. The majority (55.40%) of the school adolescents were poor academic performers. The frequency of adolescents' poor academic performance was significantly high (71.05%) among food insecure households (P < 0.001) as compared to their counterparts whose households were food

**Data Availability Statement:** The data that supports the findings of this study is available on request from the Jimma University Repository and

Health Institute Department (contact via addisalemk2009@gmail.com and ero@ju.edu.et). In addition to that, the data can also be requested from the corresponding author (contact via sagalsama24@gmail.com). The data are not publicly available due to their containing information that could compromise the privacy of research participants.

**Funding:** The author(s) received no specific funding for this work.

**Competing interests:** The authors have declared that no competing interests exist.

**Abbreviations:** AOR, Adjusted Odd Ratio; ANP, Afterschool Nutrition Programs; CI, Confidence Interval; COR, Crude Odds Ratio; HFIAS, Household Food Insecurity Access Scale; IDP, Internally Displaced Population; MoE, Ministry of Education; NSLP, National School Lunch Program; SDGs, Sustainable Development Goals; SBP, School Breakfast Program; TV, Television; VIF, Variance Inflation Factor; WHO, World health organization; WFP, World Food Programme.

secure (32.68%). On multivariable analysis, household food insecurity (AOR = 5.24, 95%CI = 3.17–8.65), school absenteeism (AOR = 3.49, 95%CI = 2.20–5.53), spending >2h/day watching TV / screen media use (AOR = 9.08, 95%CI = 4.81–17.13), high and middle wealth households (AOR = 0.51, 95%CI = 0.30–0.88) (AOR = 0.40, 95%CI = 0.21–0.76) and habitual breakfast consumption (AOR = 0.08, 95%CI = 0.03–0.20) had shown statistically significant association with academic performance among primary school adolescents.

## Conclusion

The present study revealed that household food insecurity has a high association with adolescents' academic performance. The prevalence of food insecurity is moderate, based on the household food insecurity access scale. The results indicate the need for policies and programs intended to improve household income by developing income-generation programs for lower-income families and enhance feeding programs such as national school lunch and school feeding across schools in the country.

## Background

Academic performance plays an important role in improving the quality of education, which is essential for children to be socially and economically productive for their growth. Studies showed that academic performance determines the success and failure of any educational system. Good academic performance leads to positive educational attainment. This is an important aspect in determining the quality of education in the country and the prosperity of the rest of life [1]. In the context of education, academic performance is an educational goal that a student, teacher, or institution can achieve over a period of time and is measured either by an exam or continuous evaluation [2].

Health and nutrition issues for schoolchildren and adolescents are major barriers to accessing quality education and therefore affect the achievement of the Sustainable Development Goals (SDGs) of quality education and lifelong learning opportunities for all [3]. According to research studies, food insecurity adversely affects children's health, nutrition, and academic performance by impairing students' ability to learn [4]. Food insecurity occurs when all household members do not have sufficient access to adequate quality and quantity of food to maintain an active and healthy lifestyle [5]. Children living in food-insecure households are more likely to have a poor diet, which can lead to malnutrition and a subsequent decline in academic performance [6].

Children need consistent access to sufficient food with regard to quantity and quality for optimal physical, social, emotional, and cognitive growth during each stage of life [7, 8]. Therefore, good school performance is an essential prerequisite for promoting future life opportunities in terms of employment and health [6]. Poor academic performance has a negative impact on both individuals and society, including difficulty in obtaining and understanding information, reduced income due to a low-quality occupation, reduced self-esteem, and poor health [9]. Facilitating school education, personal learning, and skill building will help students access and understand information for better health [9]. However, it is very important for children to recognize the importance of good academic performance in order to ensure the future well-being of the country [9].

Poor school involvement and attendance at school due to a lack of access to quality foods adversely affect adolescents' educational achievement, resulting in poor academic performance, repeated grades, early dropouts, late arrivals, etc. It causes anxiety, aggression, and impaired social development in schoolchildren as well, thus reducing productivity in adulthood and the next generation [10–14]. Moreover, several studies have shown that persistent deficiencies in micronutrient and macronutrient intakes are associated with impaired academic performance, social development, and an increased likelihood of being diagnosed with psychosocial disorders [10].

Numerous studies have attempted to investigate the association between food insecurity and academic achievement, and there was a negative association between food insecurity and children's school performance [3, 13, 15–18]. This indicates that it is necessary to understand the household food insecurity effect on adolescents' academic performance in the current study area. Therefore, this study is set out to assess the effect of household food insecurity on adolescents' academic performance in Hargeisa, Somaliland.

## Methods

The study was conducted in Hargeisa, the capital of Somaliland, a self-declared independent state without international recognition located in the northwestern part of Somalia. The area is located at a latitude and longitude of 11˚ 27'N and 42˚ 35'E with an elevation of 1,334 meters above sea level. It has six districts with an estimated population of 1.5 million. It has twenty-one governmental primary schools, fourteen private primary schools, eighteen kindergarten schools, and six boarding schools. The study was conducted from December 2021 to March 2022. A school-based, cross-sectional study was conducted. The source population was all students attending the governmental primary schools in Hargeisa. The study excludes private schools. This is because lower-income students attend public schools due to economic constraints on attending private schools. The study focused on public schools in order to highlight key factors that contribute to academic performance among lower-income students who ideally attend public schools over private ones. Hence, presenting evidence from governmental schools could lead policymakers to act towards empowering public schools' performance, which has a direct effect on private schools. The study population was all randomly selected students who attended governmental primary schools during the study period and fulfilled the inclusion criteria. The inclusion criteria were all students between 10–19 years of age (adolescents), those who were residents of the town for at least six months, those whose parents or caregivers consented to participate in the study, and those who were not seriously ill.

### Sample size and sampling procedure

The sample size was determined using a single population proportion formula. The assumptions considered in the determination of the sample size were 50% prevalence of students with poor academic performance since there is no previous prevalence, confidence level of 95% (1.96), margin of error of 5%, and design effect of 1.5. Based on this assumption, the calculated sample size was 576. After adding 10% for non-response, the final sample size was 634.

### Sampling techniques and procedures

Multistage sampling method was used to select participants, the sampling frame of the study schools was the list of the governmental primary schools obtained from Ministry of Education (MoE). Of the twenty-one governmental primary schools, six schools were selected randomly by using a lottery method based on WHO recommendation to include at least 30% of the total schools. In these selected schools, they serve about 7250 students attending a full primary

school cycle. The determined sample size was proportionally allocated to the selected schools and to give equal probability of being selected into the study; a simple random sampling method (lottery method) was used to select the study units from a list of students obtained from the schools.

## Data collection tools and methods

The data were collected through face-to-face interviews using an interview-administered questionnaire that was prepared in English and later translated into the Somali language. The standardized tool for measuring the wealth index was adopted from the EDHS (Ethiopia Demographic and Health Survey) in 2016 and analyzed using principal components analysis (PCA).

**Academic performance.** It is a comprehensive measure at the school level based on a student's grade score average on the standard school achievement test. Poor and good academic performance are defined as a score below and above the average of the students' grade score, in conformity with a mean score of 50% set as the promoted mark by Somaliland's national examination center. The participant's academic performance was assessed using midterm examinations' grade scores in all subjects. The mean score of academic performance was categorized and recorded as high academic performance (≥75%), average academic performance (50–74%), and low academic performance (<50%) and the mean score was further dichotomized into poor (≤49%) and good (≥50%).

**Household food insecurity.** Household food insecurity was measured in the last three months using an adaptation of the Household Food Insecurity Access Scale (HFIAS) [19]. Previously verified in many countries [11, 18, 20]. HFIAS consists of nine items asking respondents how often they experience different situations and levels of food insecurity. Each of the nine items was scored from 0 to 3, and the response options were none coded as "0," rarely as "1," sometimes as "2," and often as "3." For descriptive purposes, HFIAS was categorized into different degrees (or prevalence) of food securely, mildly, moderately, and severely food insecure [19]. However, for further analysis, food security was dichotomized into "food secure" and "food insecure," which was used constantly throughout the study.

**Socio-demographic.** The overall socio-demographic characteristics of both students and their parents were assessed, including age, sex, paternal and maternal marital status, education, and employment. House ownership type was asked of parents and dichotomized into "privately owned" and "rented house." Additionally, parents were asked whether they receive financial support or not, and responses were dichotomized into "yes" and "no."

**Wealth index.** It is a measure of the household's economic situation. The wealth index was generated using Principal Component Analysis (PCA) by considering the adolescent's household assets, such as a working TV or radio, refrigerator, house ownership, washing machine, car, bicycle, computer or laptop, electricity, table and chairs, a bed with cotton, sponge, or spring mattress, as well as a gas cooker, mobile phone, and bank account. The index had adequate internal consistency (Cronbach's alpha = 0.864). All "yes" responses were coded as one and "no" responses were coded as zero. Finally, the results were converted into tertiles and categorized into lower, moderate, and higher [21].

**Illnesses and feeding practices.** The students' overall illness and feeding practices were assessed. Responses to breakfast were dichotomized into three groups as follows: "having breakfast rarely (0–2 days)" or "having breakfast occasionally (3–4 days)" or "having breakfast frequently (5–7 days)." We also asked for information on the meal frequency, and responses were dichotomized into "eating meals less than 3 times per day" or "eating meals equal to or more than 3 times per day." The students' illnesses were assessed, asking "whether they were

ill, had diarrhea, fever, or cough for the past 4 weeks." Responses were dichotomized into "yes" or "no."

**School absenteeism.**   Absence was defined as any illegal absence from school of at least one day in the previous semester, excluding days when school is officially closed (national or religious holidays) [3]. The students were asked, "How many days have you been absent from the school during the first term?" Then responses were categorized into "never absent," "absent 1–3 days," "4 days & above," Then, the absenteeism was further dichotomized into "absent" and "never absent."

**Study time.**   The study duration at home was divided into "almost none," "1 to 2 h," "2 to 3 h," "3 to <4 h," and "≥4." Then, the study duration was dichotomized into "<1h" or "≥ 1 h."

**Physical activity.**   Vigorous physical activity was assessed because it is strongly and independently associated with markers of cardio-metabolic health and can be more reliable than light or moderate physical activity. Students were asked if they were engaged in intense activity for more than three days a week, and the "yes" answer was coded (1) and the "no" answer was coded (0). Intense activity was defined as an exercise, game, or dance that made them breathe hard, made their legs feel tired, or made them sweat [22].

**Screen time** is the duration that children or adolescents spend on television viewing, video game playing, and Internet use. The students were asked "the average number of hours they spent in a day watching TV, movies, video game playing, mobile phones, Internet, and overall screen media" [23]. This was separated from the weekend and the time of their homework, which was not included in our assessment of screen time use. Response options were as follows: None, 1 h/day, 1–2 h/day, 3–4 h/day, 5–6 h/day, and 7 h/day. Finally, responses were dichotomized into "≤ 2 h/day" and ">2h/day."

## Data processing and analysis processing procedures

The data were entered into Epi Data Version 3.1 and then exported to Statistical Package for the Social Sciences (SPSS) Version 26 for statistical analysis. Data summary statistics were computed to define the study participants in relation to important variables.

The bivariate analysis was performed to select candidate variables for the multivariable regression analysis at P≤0.25, and Crude Odd Ratio (COR) with 95% CI was also used. Accordingly, variables including food security status, student's birth order, head of household, maternal education, paternal education, household wealth index, habitual breakfast intake, daily meal frequency, school absenteeism, study duration at home, screen time, and hand wash practices were entered into a multivariable logistic regression to identify significant variables and control the potential confounding effects. The final statistical association was measured by AOR and 95% CI. The goodness of the model was tested by the Hosmer-Lemeshow test, and the model was fit with P -value = 0.600. Principal Component Analysis (PCA) was done to generate a household wealth index score. All assumptions of PCA were checked and fulfilled; the final factors were taken and categorized into three categories: "lower," "moderate," and "higher."

## Data quality management

A structured questionnaire was translated to Somali and back to English for consistency. Three days of intensive training were given to data collectors and supervisors. Written consent was obtained from participants and their guardians or parents. Telephone interviews were used for parents or guardians who were unable to attend school during data collection (due to childcare needs, illness, work restrictions, etc.).

A pre-test was conducted at five percent of the whole sample size in Arabsiyo, a nearby primary school outside the study area one week before the data collection time. The multicollinearity of independent variables was checked using Variance Inflation Factor (VIF) < 10 in the regression model.

### Ethics approval and consent to participate

Ethical clearance was obtained from the ethical review committee of Jimma University. Official permission was acquired from the MoE in Somaliland. Formal letters of cooperation were written to all selected public primary schools' administrations in Hargeisa City. Verbal and written consent was obtained from participants and their parents or caregivers. Privacy, anonymity, and confidentiality were ensured throughout the process of the study. Finally, this study was conducted in accordance with the declaration of Sagal Mohamed Adam.

## Results

### Socio-demographic and economic characteristics

A total of six hundred thirty-four school adolescents were planned to be involved in the study, and six hundred thirty were recruited in this study with a 99.3% response rate. The mean age of the respondents was 14.04 (±1.69), and more than half (58%) of the respondents were in the age range of 13–15 years. The majority of study participants were female (51.3%). Most of the households are composed of more than six members (62.9%). As a result, students whose mothers and fathers had no formal education had poorly attained academic performance (50.3% and 16.3%) compared to those who had good academic performance (26.5% and 7.9%, respectively). Half of students (59.9%) from the lower household wealth index had poor academic performance compared to their peers, who had good academic performance (37.7%) at P<0.001 "Table 1."

### Academic performance and household food insecurity

The frequency of poor academic performance is significantly higher in adolescents from food-insecure households than in adolescents from food-secure households (P<0.001).

The majority (71%) of adolescents from food-insecure households had poorer academic performance than adolescents from food-secure households (32.7%) "Fig 1."

### Adolescent's illness, feeding, lifestyle and environmental health related factors

The majority of participants, 410 (65.1%), did not eat breakfast before going to school, and this shows that students who did not eat their breakfast performed poorly (66.5%) compared to those who took their morning breakfast (33.5%). Of the total participants, 339 (53.8%) were absent from school, and those who were absent had poor academic performance (70.8%) compared to their peers who were not absent from school (29.2%). Three hundred forty-six (55%) respondents were spending over 2 hours per day on screen watching, including TV watching or video game playing, and 61.9% of them had poor academic performance compared to those who watched less than 2 hours per day (38.1%). The majority of 366 (58.1%) study participants drank water from tap water sources. Three hundred fifty-nine (56.2%) of respondents did not wash their hands with soap after the toilet; of those, 62.5% had poor academic performance compared to their counterparts who wash their hands with soap after toilet "Table 2."

**Table 1. Socio-demographic and economic characteristics of school adolescents and their families by academic performance in Hargeisa city, Somaliland.** (n = 630) 2022.

| Characteristics | Category | Good AP n(%) | Poor AP n(%) | % | P |
|---|---|---|---|---|---|
| Student age (years) | 10–12 | 60 (21.4%) | 72 (20.6%) | 21.0 | 0.361 |
| | 13–15 | 170 (60.5%) | 200 (57.3%) | 58.7 | 0.232 |
| | 16–19 | 51 (18.1%) | 77 (22.1%) | 20.3 | |
| Gender | Male | 132 (47.0%) | 175 (50.1%) | 48.7 | 0.429 |
| | Female | 149 (53.0%) | 174 (49.9%) | 51.3 | |
| Student birth Order | First child | 78 (27.8%) | 82 (23.5%) | 25.4 | |
| | Second child | 90 (32.0%) | 92 (26.4%) | 28.9 | 0.897 |
| | Third &above child | 113 (40.2%) | 175 (50.1%) | 45.7 | 0.052 |
| Household head | Father | 142 (50.5%) | 166 (47.6%) | 48.9 | 0.002 |
| | Mother | 104 (37.0%) | 98 (28.1%) | 32.1 | 0.000 |
| | Other family member | 35 (12.5%) | 85 (24.4%) | 19 | |
| Household head's gender | Male | 161 (57.3%) | 205 (58.7%) | 58.1 | |
| | Female | 120 (42.7%) | 144 (41.3%) | 41.9 | 0.715 |
| Parent/caregiver's Marital status | Married | 202 (71.9%) | 247 (70.8%) | 67.8 | 0.759 |
| | Unmarried [b] | 79 (28.1%) | 102 (29.2%) | 32.2 | |
| Mother's education | No formal education | 74 (26.5%) | 174 (50.3%) | 39.7 | 0.000 |
| | Read and write [a] | 74 (26.5%) | 61 (17.6%) | 21.6 | 0.229 |
| | Primary level | 53 (19.0%) | 50 (14.5%) | 16.5 | 0.131 |
| | Secondary level | 53 (19.0%) | 48 (13.9%) | 16.2 | 0.161 |
| | Collage and above | 25 (9.0%) | 13 (3.8%) | 6.1 | |
| Mother's occupation | Housewife | 180 (64.5%) | 237 (68.5%) | 66.7 | 0.526 |
| | Government employee | 9 (3.2%) | 12 (3.5%) | 3.4 | 0.759 |
| | Merchant | 37 (13.3%) | 36 (10.4%) | 11.7 | 0.576 |
| | Others [c] | 53 (19.0%) | 61 (17.6%) | 18.3 | |
| Father's education | No formal education | 22 (7.9%) | 57 (16.6%) | 12.7 | .000 |
| | Read and write [a] | 30 (10.8%) | 43 (12.5%) | 11.7 | .046 |
| | Primary level | 40 (14.3%) | 57 (16.6%) | 15.6 | .028 |
| | Secondary level | 44 (15.8%) | 67 (19.5%) | 17.8 | .010 |
| | Collage and above | 143 (51.3%) | 120 (34.9%) | 42.2 | |
| Father's occupation | Government employee | 105 (37.6%) | 126 (36.6%) | 37.1 | 0.724 |
| | Merchant | 68 (24.4%) | 88 (25.6%) | 25.0 | 0.624 |
| | Others [c] | 98 (35.1%) | 122 (35.5%) | 35.3 | 0.672 |
| | Unemployed | 8 (2.9%) | 8 (2.3%) | 2.6 | |
| Household size | ≤6 | 107 (38.1%) | 127 (36.4%) | 37.1 | 0.663 |
| | >6 | 174 (61.9%) | 222 (63.6%) | 62.9 | |
| House ownership type | Private | 179 (63.7%) | 209 (59.9%) | 61.6 | 0.328 |
| | Rent | 102 (36.3%) | 140 (40.1%) | 38.4 | |
| Household wealth Index (tertiles) | Lower | 106 (37.7%) | 209 (59.9%) | 50.0 | 0.000 |
| | Middle | 64 (22.8%) | 56 (16.0%) | 19.0 | 0.534 |
| | Higher | 111 (39.5%) | 84 (24.1%) | 31.0 | |
| Financial support | No | 159 (56.6%) | 207 (59.3%) | 58.1 | 0.490 |
| | Yes | 122 (43.4%) | 142 (40.7%) | 41.9 | |

[a] who learnt adult education or attended non-formal education

[b] single, divorced, separated, or widowed

[c] non-government employee or daily laborer.

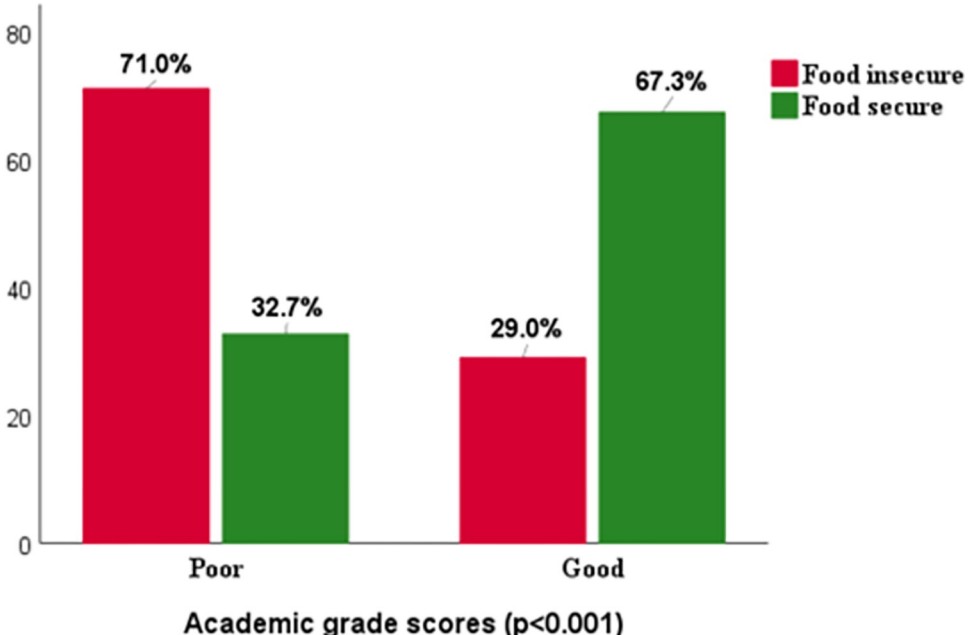

**Fig 1. AP by FI.**

## Academic performance

The average grade score of the study participants was 52.6 ± 17.6, with a minimum and maximum value of 17.40% and 89.70%, respectively. Ninety-six (15.2%) had high academic performance (≥75%), and 185 (29.4%) had average academic performance (50–74%). The majority of the participants, 349 (55.4%) had low academic performance [<50%]. "Fig 2."

## Predictors of academic performance

Findings of multivariable logistic regression analysis showed that household food insecurity (AOR = 5.24, 95%CI = 3.17–8.65), school absenteeism (AOR = 3.49, 95%CI = 2.20–5.53), spending >2h/day watching TV or screen media use (AOR = 9.08, 95%CI = 4.81–17.13), high and middle wealth households (AOR = 0.51, 95%CI = 0.30–0.88), (AOR = 0.40, 95%CI = 0.21–0.76) and habitual breakfast intake (AOR = 0.08, 95%CI = 0.03–0.20) were significantly associated with academic performance among school adolescents "Table 3."

## Discussion

The overall prevalence of household food insecurity among school adolescents in Hargeisa City was 59.21%. This finding is much higher than the national prevalence of household food insecurity, which is 53% [24]. This difference may be attributed to the use of different methodologies, including sampling techniques, in these studies. Furthermore, the prevalence of poor academic performance among primary school adolescents in this study was 55.4%. This prevalence is higher than the national prevalence of poor academic performance among primary school students, which was 51.97% in 2018 [25].

The present study revealed that poor academic performance among school adolescents was significantly higher in food-insecure households. Adolescents from food-insecure households had lower academic scores than those from food-secure households. This finding is consistent with previous studies [6–8, 13]. This may be attributed to poor understanding of the lecture,

**Table 2. Student's illness, feeding, lifestyle and environmental health related characteristics of school adolescents by academic performance in Hargeisa city, Somaliland.** (n = 630) 2022.

| Characteristics | Category | Good AP n(%) | Poor AP n(%) | % | P |
|---|---|---|---|---|---|
| Any illness in the last 4 weeks | No | 166 (59.1%) | 208 (59.6%) | 59.4 | 0.894 |
| | Yes | 115 (40.9%) | 141 (40.4%) | 40.6 | |
| Diarrhea in the last 4 weeks | No | 263 (93.6%) | 320 (91.7%) | 92.5 | 0.367 |
| | Yes | 18 (6.4%) | 29 (8.3%) | 7.5 | |
| Fever/cough illness in the last 4 weeks | No | 156 (55.5%) | 208 (59.6%) | 57.8 | 0.303 |
| | Yes | 125 (44.5%) | 141 (40.4%) | 42.2 | |
| Breakfast intake | No | 111 (39.5%) | 232 (66.5%) | 65.1 | 0.000 |
| | Yes | 170 (60.5%) | 117 (33.5%) | 34.9 | |
| Habitual breakfast intake | Rare (0–2 days) | 21 (7.5%) | 79 (22.6%) | 15.9 | 0.000 |
| | Occasional (3–4 days) | 113 (40.2%) | 159 (45.6%) | 26.3 | 0.000 |
| | Frequent (5–7 days) | 147 (52.3%) | 111 (31.8%) | 57.8 | |
| Daily meal frequency | <3 times | 108 (38.4%) | 157 (45.0%) | 42.1 | 0.098 |
| | ≥3 times | 173 (61.6%) | 192 (55.0%) | 57.9 | |
| Absenteeism from school | Absent | 92 (32.7%) | 247 (70.8%) | 53.8 | 0.000 |
| | Never absent | 189 (67.3%) | 102 (29.2%) | 46.2 | |
| Study duration at home | <1 h/day | 108 (38.4%) | 217 (62.2%) | 51.5 | 0.000 |
| | ≥1 h/day | 173 (61.6%) | 132 (37.8%) | 48.5 | |
| Typical nightly sleep | <8 h | 113 (40.2%) | 127 (36.4%) | 38.1 | 0.326 |
| | ≥8 h | 168 (59.8%) | 222 (63.6%) | 61.9 | |
| Vigorous physical activity (PA) | No | 206 (73.3%) | 242 (69.3%) | 71.1 | 0.275 |
| | Yes | 75 (26.7%) | 107 (30.7%) | 28.9 | |
| Smoking status | Non-daily smokers | 256 (91.1%) | 312 (89.4%) | 90.2 | 0.476 |
| | Daily smokers | 25 (8.9%) | 37 (10.6%) | 9.8 | |
| Screen time on weekday | <2 h/day | 151 (53.7%) | 133 (38.1%) | 45.0 | |
| | ≥2h/day | 130 (46.3%) | 216 (61.9%) | 55.0 | 0.000 |
| Source of drinking water | Protected spring/ well | 124 (44.1%) | 140 (40.1%) | 41.9 | 0.310 |
| | Tap water | 157 (55.9%) | 209 (59.9%) | 58.1 | |
| Adolescent's hand washing practice with soap after toilet | No | 141 (50.2%) | 218 (62.5%) | 56.2 | 0.002 |
| | Yes | 140 (49.8%) | 131 (37.5%) | 43.8 | |
| Adolescent's hand washing practice with soap before meal | No | 151 (53.7%) | 194 (55.6%) | 54.8 | 0.643 |
| | Yes | 130 (46.3%) | 155 (44.4%) | 45.2 | |

tiredness, poor attendance due to the inability to purchase food to eat at school, lack of concentration in class, and thinking about food instead of studying. Various studies reported that respondents with food insecurity experience poor eating habits and inadequate energy, which can lead to physical fatigue and poor concentration in the class [26–28]. Food insecurity has also been shown to lead to malnutrition, undermining outcomes that may contribute to poor academic performance in children, such as poor dietary intake, poor psychosocial outcomes, mental health, and cognitive development [6, 14, 18]. However, a multisectoral approach to enhancing students' access to food at the school level is required. Although the WFP school feeding program is only active at public schools for Internally Displaced Populations (IDPs), this may not be effective for the rest of the food-insecure students studying at other public schools in the city.

Our results imply that having a lower household wealth index has a negative impact on academic performance. This indicates that adolescents from lower wealth households had poor academic performance as compared to their peers from high- and middle-wealth households.

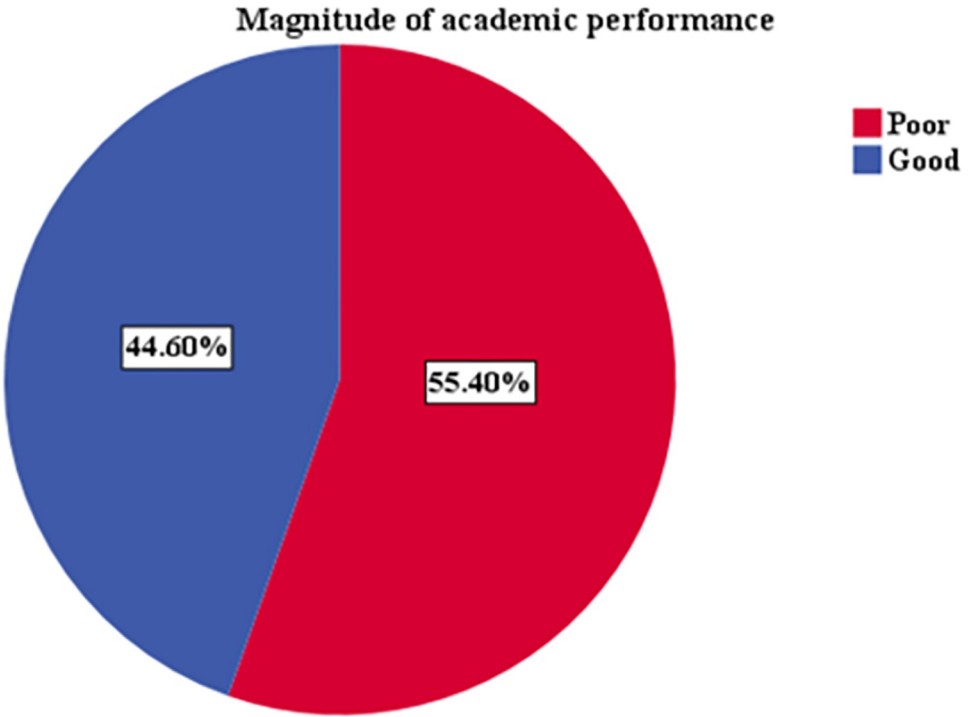

**Fig 2. Pie chart AP.**

This finding has consistency with previous studies done in Malaysia and Iran [29, 30]. This may be due to a lack of access to high-quality foods, balanced nutritional intake, health services, and additional learning materials that contribute to school participation and learning abilities. Plenty of research studies support the idea that households with better economic status can usually provide a better educational basement for their children, while those from low-income households are more likely to be retained in grade, have health problems, poor nutrition, and poor academic performance than their counterparts from higher-income households [31–33].

The study also revealed that screen time has a significant association with academic performance. Adolescents who spend ≥2 hours per day on watching TV, video game playing, and overall screen media use had lower academic scores than those who spent < 2 hours per day. The findings of this study are consistent with previous research studies. A systematic review and meta-analysis showed that there is a strong association between academic performance and screen time [33–36]. This could be due to a loss of attention in the class, lack of interest and spending less time on activities that can contribute to good academic achievements, such as reading, doing homework, and regular studying. A research study found that screen time has a detrimental effect on school performance because it eliminates the time that would normally be spent doing schoolwork, reading for pleasure, or engaging in other educational activities [34]. This finding supports the 24-hour movement guidelines in terms of screen time, which suggests adherence to screen time recommendations is associated with good academic performance among school adolescents [23].

Habitual consumption of breakfast has a significant association with academic performance. Students who consumed their breakfast frequently had good academic performance as compared to their counterparts. This finding is similar to previous studies reporting an association between breakfast consumption and academic performance [37–39]. This may be for

**Table 3. Multivariable logistic regression analysis of factors associated with academic performance among primary school adolescents in Hargeisa city, Somaliland (n = 630) 2022.**

| Variables | Categories | Academic Performance | | COR (95% CI) | AOR (95% CI) | P-value |
|---|---|---|---|---|---|---|
| | | Poor (%) | Good (%) | | | |
| Student birth order | First child | 82 (51.2) | 78 (48.8) | 1 | 1 | |
| | Second/ above | 267 (56.8) | 203 (43.2) | 1.251(0.873, 1.793) | 1.254(0.748, 2.102) | 0.391 |
| Household head | Father | 166 (53.9) | 142 (46.1) | 0.481(0.306, 0.757) * | 0.535(0.254, 1.126) | 0.099 |
| | Mother | 98 (48.5) | 104 (51.5) | 0.388(0.240, 0.627) ** | 0.529(0.242, 1.156) | 0.110 |
| | Relatives | 85 (70.8) | 35 (29.2) | 1 | 1 | |
| Maternal education | No education | 174 (70.2) | 74 (29.8) | 4.522(2.194, 9.320) ** | 2.473(0.906, 6.745) | 0.077 |
| | Read and write [a] | 61 (45.2) | 74 (54.8) | 1.585(0.748, 3.359) | 0.403(0.139, 1.167) | 0.094 |
| | Primary level | 50 (48.5) | 53 (51.5) | 1.814(0.837, 3.932) | 1.009(0.360, 2.829) | 0.986 |
| | Secondary level | 48 (47.5) | 53 (52.5) | 1.742(0.802, 3.783) | 0.416(0.147, 1.179) | 0.099 |
| | Collage & above | 13 (34.2) | 25 (65.8) | 1 | 1 | |
| Paternal education | No education | 57 (72.2) | 22 (27.8) | 3.087(1.784, 5.344) ** | 1.028(0.452, 2.336) | 0.947 |
| | Read and write [a] | 43 (58.9) | 30 (41.1) | 1.708(1.010, 2.889) * | 1.354(0.629, 2.918) | 0.439 |
| | Primary level | 57 (58.8) | 40 (41.2) | 1.698(1.060, 2.721) * | 1.606(0.774, 3.333) | 0.203 |
| | Secondary level | 67 (60.4) | 44 (39.6) | 1.815(1.156, 2.849) * | 1.676(0.885, 3.173) | 0.113 |
| | Collage & above | 120 (45.6) | 143 (54.4) | 1 | 1 | |
| Food secure status | Food insecure | 265 (71.0) | 108 (29.0) | 5.053(3.584, 7.125) ** | 5.245(3.178, 8.654) | 0.000 |
| | Food secure | 84 (32.7) | 173 (67.3) | 1 | 1 | |
| Household wealth index | Lower | 209 (66.3) | 106 (33.7) | 1 | 1 | |
| | Middle | 56 (46.7) | 64 (53.3) | 0.444(0.289, 0.681) ** | 0.403(0.212, 0.767) | 0.006 |
| | Higher | 84 (43.1) | 111 (56.9) | 0.384(0.266, 0.554) ** | 0.515(0.301, 0.883) | 0.016 |
| Habitual breakfast intake | Rarely (0-2d) | 79 (79.0) | 21 (21.0) | 1 | 1 | |
| | Occasional(3-4d) | 124 (74.7) | 42 (25.3) | 0.785(0.433, 1.423) | 0.522(0.219, 1.246) | 0.143 |
| | Frequency (5-7d) | 146 (40.1) | 218 (59.9) | 0.178(0.105, 0.301) ** | 0.080(0.031, 0.205) | 0.000 |
| Daily meal frequency | <3 times | 157 (59.2) | 108 (40.8) | 1.310(0.951, 1.803) | 0.719(0.423, 1.223) | 0.224 |
| | ≥3 times | 192 (52.6) | 173 (47.4) | 1 | 1 | |
| School absenteeism | Absent | 247 (72.9) | 92 (27.1) | 4.975(3.542, 6.986) ** | 3.493(2.205, 5.534) | 0.000 |
| | Never absent | 102 (35.1) | 189 (64.9) | 1 | 1 | |
| Study time at home | <1 h/day | 85 (68.5) | 39 (31.5) | 1.998(1.316, 3.032) * | 1.149(0.598, 2.210) | 0.676 |
| | ≥1 h/day | 264 (52.2) | 242 (47.8) | 1 | 1 | |
| Screen time | < 2 h/day | 133 (40.5) | 195 (59.5) | 1 | 1 | |
| | ≥2h/day | 216 (71.5) | 86 (28.5) | 3.682(2.639, 5.139) ** | 9.087(4.819, 17.13) | 0.000 |
| Hand wash practice after toilet | No | 213 (60.2) | 141 (39.8) | 1.555(1.132, 2.137) * | 0.660(0.368, 1.183) | 0.163 |
| | Yes | 136 (49.3) | 140 (50.7) | 1 | 1 | |

COR Crude Odd Ratio, AOR Adjusted Odd Ratio, CI Confidence Interval

*Significant at <0.05

**Significant at <0.001

[a]who learnt adult education or attended non-formal education.

those who have taken their breakfast habitually have sufficient energy to be active, attentive, participate in class, walk long distances to school, concentrate, and remain engaged while in school. A cross-sectional study conducted in southern Ethiopia also showed that breakfast meals contribute to improving cognition [40]. Children who consume breakfast are more likely to meet their energy and overall nutrient requirements compared to those who do not have breakfast [41].

School absenteeism was another factor that was significantly associated with academic performance. Adolescents who were absent from school had poor academic performance as compared to their counterparts. This finding is consistent with previous research studies [42, 43]. This might be due to failure to catch the valuable information from the lectures and the unclear concepts that lead to inadequate learning that may result from inadequate access to food and enough energy. Adolescents may skip school because they cannot afford food to attend school or because they do not have enough food at home to get enough energy to walk long distances to school [18]. However, research studies have shown that regular school attendance promotes teamwork, self-confidence, and the ability to understand basic conceptual learning [43].

Ensuring food security for students in school is very important, considering that this is the environment where children eat most of their meals. Food consumption among students is not yet assessed, which is a crucial indicator for the assessment of food insecurity. Thus, a more detailed analysis covering different aspects of food consumption would allow for a more comprehensive understanding of students' eating habits. The association of food security status with academic performance among adolescents may differ in different communities. Therefore, the findings may serve to provide evidence regarding the relationship between food insecurity and academic performance, specifically in developing countries that were examined in a few studies. Again, the influence of sociodemographic factors, including maternal education and the dynamics of female-headed households in relation to food security, is required to be studied deeply. Moreover, to explore clear associations between food insecurity and developmental consequences, studies and new approaches are needed. For instance, measuring the impact of national programs such as the School Feeding Program in Somaliland on children's academic performance can serve as an opportunity to explore such associations. In any case, the results suggest the need for nutrition and educational supplemental programs for low-income and food-insecure students to provide them with equal opportunity.

The current study had limitations. The cross-sectional design cannot establish a causal relationship between variables. The other limitation of this study is that it was conducted in public primary schools, which may not be representative of private schools. There might also be recall bias as respondents may forget past food intake, but data collectors were given intensive training on how to probe respondents to remember their food intake. Another limitation was the lack of similar studies for making comparative discussions, especially among school adolescents in Hargeisa. Further study in this area should be conducted to support these findings. Despite these, we utilized an adequate sample size and the mean score of the students' average as the cut-off value to say good or poor academic performance, which could help the findings be generalizable to primary school students.

## Conclusion

This study revealed the prevalence of household food insecurity among school adolescents in Hargeisa City was 59.21%, and household food insecurity was strongly associated with poor academic performance. School absenteeism, high and middle-income households, habitual breakfast intake, prolonged television watching, or screen media use also showed a significant association with academic performance.

Therefore, a multi-sectoral response is needed to develop policies and strengthen programs intended to improve household income through income generation approaches for lower-income households. In addition to that, enhancing school feeding programs that are currently specific to a small number of schools, effectively introducing school gardening programs or including food gardening at school in the education curriculum across the country, initiation

of the School Breakfast Program (SBP), and Afterschool Nutrition Programs (ANP) in order to address food insecurity and its negative consequences on education. Although the WFP school feeding program is only active at public schools for Internally Displaced Populations (IDPs), this may not be effective for the rest of the food-insecure students studying at other public schools in the city.

## Supporting information

**S1 File.**
(DOCX)

## Acknowledgments

I would like to express my sincere gratitude to Jimma University, the Institute of Health, data collectors, supervisors, and the Department of Nutrition and Dietetics for giving us this chance, and also to the Somaliland Ministry of Education, governmental schools, and study subjects for their respected involvement in the insight of this study.

## Author Contributions

**Conceptualization:** Sagal Mohamed Adam, Melese Sinaga Teshoma, Dessalegn Tamiru.

**Data curation:** Sagal Mohamed Adam.

**Formal analysis:** Sagal Mohamed Adam.

**Investigation:** Sagal Mohamed Adam.

**Methodology:** Sagal Mohamed Adam, Melese Sinaga Teshoma, Dessalegn Tamiru.

**Software:** Sagal Mohamed Adam.

**Supervision:** Melese Sinaga Teshoma, Dessalegn Tamiru.

**Validation:** Sagal Mohamed Adam, Melese Sinaga Teshoma, Dessalegn Tamiru.

**Visualization:** Sagal Mohamed Adam, Melese Sinaga Teshoma, Dessalegn Tamiru.

**Writing – original draft:** Sagal Mohamed Adam.

**Writing – review & editing:** Sagal Mohamed Adam, Melese Sinaga Teshoma, Awale Sh. Dahir Ahmed, Dessalegn Tamiru.

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
