## [Decision Letter · Decision Letter 0]

27 Dec 2023

PONE-D-23-31840Household food insecurity and its association with academic performance among primary school adolescents in Hargeisa City, SomalilandPLOS ONE

Dear Dr. Adam,

Thank you for submitting your manuscript to PLOS ONE. After careful consideration, we feel that it has merit but does not fully meet PLOS ONE’s publication criteria as it currently stands. Therefore, we invite you to submit a revised version of the manuscript that addresses the points raised during the review process.

We look forward to receiving your revised manuscript.

Kind regards,

Larissa Loures Mendes, Ph.D.

Academic Editor

PLOS ONE

A clean copy of the edited manuscript (uploaded as the new *manuscript* file).

4. In the online submission form you indicate that your data is not available for proprietary reasons and have provided a contact point for accessing this data. Please note that your current contact point is a co-author on this manuscript. According to our Data Policy, the contact point must not be an author on the manuscript and must be an institutional contact, ideally not an individual. Please revise your data statement to a non-author institutional point of contact, such as a data access or ethics committee, and send this to us via return email. Please also include contact information for the third party organization, and please include the full citation of where the data can be found.

Additional Editor Comments:

Dear Authors,

Thank you for the opportunity to edit this manuscript. After the peer review, I realized that several modifications to the manuscript will be necessary in order to consider it for publication. I would like you to consider the reviewers' suggestions and questions and resubmit the manuscript after the modifications.

Reviewers' comments:

Reviewer's Responses to Questions

**Comments to the Author**

1. Is the manuscript technically sound, and do the data support the conclusions?

Reviewer #1: Yes

Reviewer #2: Yes

2. Has the statistical analysis been performed appropriately and rigorously? 

Reviewer #1: No

Reviewer #2: Yes

3. Have the authors made all data underlying the findings in their manuscript fully available?

Reviewer #1: Yes

Reviewer #2: Yes

4. Is the manuscript presented in an intelligible fashion and written in standard English?

Reviewer #1: No

Reviewer #2: Yes

5. Review Comments to the Author

Reviewer #1: The aim of this manuscript was to assess the effect of household food insecurity on adolescents’ academic performance in Hargeisa, Somaliland. This is an interesting topic, but several aspects of the text need to be improved.

Comments

I recommend that a native speaker of English proofread the text. This will help a lot with sentence construction and improve the flow of the text. I also recommend reading the instructions for authors on the journal's formatting rules.

Methods

The results of the PCA should be presented in a supplementary material file.

You describe only the Academic performance and Household food insecurity variables. What about the other variables used in the study? For example, all the variables shown in Tables 1 and 2 should be included in the methodology and their classifications described.

Results

Include the sample number of your study and the year of collection in the tables and figures headings so that the table is complete and can be read independently of the text.

It would be interesting to present the descriptive results also stratified by academic performance (Tables 1, 2 and 3).

The main points of the tables have yet to be highlighted.

The pictures are of poor quality.

In the methods, you say that you have dichotomized the academic performance variable into good and poor, but in Figure 2 you present three categories. Standardize this.

Pay attention to the maximum number of tables and figures allowed by the journal.

Are all the variables, tables, and figures essential for your article? I suggest reviewing the data presented, as much information is not described in the methodology and variables other than those mentioned in the objective. Making the text more direct, informative, and concise would be best.

Discussion

In the discussion, you should focus on discussing the results you found, bringing in a bit of the context of the city and population evaluated, and not just comparing them with the results of other studies.

In addition, you should bring up the implications of the results. Where is the proper reflection on these results? What can the government do to improve this?

The limitations should be included in the discussion session. I also recommend presenting the strengths of the study.

The second paragraph of the conclusion should be better worked into the discussion. Including regional aspects, what already exists in Hargeisa that can help in this scenario?

Reviewer #2: Dear authors, the study presents important results for understanding the impact of food insecurity on academic performance and has significant implications for public health. However, there are some points that can be improved. The suggestions below aim to further improve the quality and impact of the article.

Methods:

Sample: The exclusion of private schools from the sample must be justified. In the methods section of the article, the author should clearly explain why they decided to exclude private schools.

In the “Sample size and sampling procedure” session, it was not clear why we chose to work with students whose performance is 50%. I suggest requires a clearer explanation of these selection criteria.

Describe what the acronym COR means in the section “Data processing analysis and analysis processing procedures”

The results included data on the adolescents' lifestyle habits, information related to basic sanitation and hygiene, as well as data on other family members. However, this information was not presented in the methods. It is important to include a comprehensive description of these aspects in the methods, ensuring consistency in the presentation of methodological procedures and allowing a more complete understanding of the approach adopted.

Results:

In the method, the authors chose to categorize data related to food security and insecurity in a dichotomous way. However, the results show the presentation of both this categorization and the previous categories. I recommend that authors clearly establish which categories will be used consistently throughout the study: whether they will be Food security and Food insecurity, or whether a more detailed approach will be applied, classifying into Food security, Mild food insecurity, Moderate food insecurity and Severe food insecurity, providing greater clarity and uniformity in the analysis and interpretation of data.

Statistical analyzes were carried out to investigate the association between food insecurity, academic performance and the data presented in the categories “Adolescent illness and factors related to food” “Factors related to adolescent lifestyle” “Environmental sanitation and factors related to hygiene”? It is unclear how the descriptive analysis of these data, as presented in the article, contributes to the study. I suggest considering excluding these data from the survey or, alternatively, providing statistical analyzes that demonstrate the relationship (or lack of relationship) between these factors and food insecurity and academic performance. This review may enhance the relevance and impact of the findings in the study.

The excerpt below is part of the methods, I suggest describing our methods and extracting the results.

“Bivariate analysis was performed to select candidate variables for multivariate regression analysis at P≤0.25. Thus, food security status, birth order, head of household, maternal education, paternal education, family wealth index, habitual breakfast intake, daily meal frequency, school absenteeism, home study duration, screen time, and practice of washing hands after using the bathroom were selected as candidate variables.”

Discussion:

Based on the results of the study, it is essential to enrich the discussion by incorporating some unaddressed points, which could add depth to the understanding of the topic. Suggest including in the discussion:

- The importance and central role of the school in ensuring food security for students, taking into account that this is the environment where children eat most of their meals.

- A more in-depth analysis of the results, including data on the family, the influence of maternal education and the dynamics of female-headed households in relation to food security.

- A more detailed analysis of food consumption is recommended in the discussion, considering that this indicator plays a fundamental role in the assessment of food security. Note a limited approach to the discussion of food consumption, with only one paragraph dedicated to the debate about breakfast. Expanding the discussion to cover different aspects of food consumption would allow for a more comprehensive understanding of students' eating habits.

Describe the limitations as a discussion paragraph and add the strengths of the study.

6. PLOS authors have the option to publish the peer review history of their article (what does this mean?). If published, this will include your full peer review and any attached files.

Reviewer #1: No

Reviewer #2: **Yes: **Nayhanne Gomes Cordeiro

---

## [Author Response · Author response to Decision Letter 0]

9 Feb 2024

Responses to peer reviewers' comments

Dear Editors and Reviewers, 

Thank you for giving me the opportunity to submit a revised draft of my manuscript titled “Household food insecurity and its association with academic performance among primary school adolescents in Hargeisa City, Somaliland” to PLOS One. We appreciate the time and effort that you and the reviewers have dedicated to providing your valuable feedback on my manuscript. We are grateful to the reviewers for their insightful comments on my paper. We have been able to incorporate changes to reflect most of the suggestions provided by the reviewers. We have highlighted the changes within the manuscript. 

Here is a point-by-point response to the reviewers’ comments and concerns. 

Comments from Reviewer 1 

Methods 

1. Comments: [The results of the PCA should be presented in a supplementary material file.]. Response: Thank you for pointing this out. We agree with this comment. Therefore, we have uploaded the supplementary file for the PCA results. 

2. Comments: [You describe only the Academic performance and Household food insecurity variables. What about the other variables used in the study? For example, all the variables shown in Tables 1 and 2 should be included in the methodology and their classifications described]. Response: Agree. We have, accordingly, included other variables descriptions. Page number (6 & 7)

Results 

1. Comments: [Include the sample number of your study and the year of collection in the tables and figures headings so that the table is complete and can be read independently of the text. It would be interesting to present the descriptive results also stratified by academic performance (Tables 1, 2 and 3). The main points of the tables have yet to be highlighted. The pictures are of poor quality. In the methods, you say that you have dichotomized the academic performance variable into good and poor, but in Figure 2 you present three categories. Standardize this. Pay attention to the maximum number of tables and figures allowed by the journal.]. Response: We agree with this and have incorporated your suggestions throughout the manuscript. We have reduced number of tables and figures previously presented. Also, we presented data stratified by academic performance. Resolved pictures quality. 

2. Comments: [Are all the variables, tables, and figures essential for your article? I suggest reviewing the data presented, as much information is not described in the methodology and variables other than those mentioned in the objective. Making the text more direct, informative, and concise would be best.]. Response: You have raised an important point here. However, we believe that most of the variables would be more appropriate because academic performance among adolescents can be impacted by other pathways. Thus, we implied with your suggestion throughout the manuscript, and we have described the required information in the methodology. 

Discussion

1. Comments: [In the discussion, you should focus on discussing the results you found, bringing in a bit of the context of the city and population evaluated, and not just comparing them with the results of other studies. In addition, you should bring up the implications of the results. Where is the proper reflection on these results? What can the government do to improve this?]. Response: Thank you for pointing this out. We agree with this comment. We have incorporated your suggestions throughout the manuscript. 

2. Comments: [The limitations should be included in the discussion session. I also recommend presenting the strengths of the study. The second paragraph of the conclusion should be better worked into the discussion. Including regional aspects, what already exists in Hargeisa that can help in this scenario?]. Response: Thank you for pointing this out. We agree with this comment. We have incorporated your suggestions throughout the manuscript. 

Comments from Reviewer 2 (Nayhanne Gomes Cordeiro)

Methods 

1. Comments: [The exclusion of private schools from the sample must be justified. In the methods section of the article, the author should clearly explain why they decided to exclude private schools]. Response: You have raised an important point here. However, we believe that exclusion of private schools from the study is our limitation. The study excludes private schools this is because lower income students attend public schools due to economical constraints in attending private schools. The study focused on public schools in order to highlight key factors that contribute to academic performance among lower income students who ideally attend public schools over private ones. Hence presenting evidence from governmental schools could lead to policy makers to take an action towards empowering public schools’ performance that has direct effect to private schools. However, including private schools would be more appropriate because it will help on generalization of our study. 

2. Comments: [In the “Sample size and sampling procedure” session, it was not clear why we chose to work with students whose performance is 50%. I suggest requires a clearer explanation of these selection criteria]. Response: Thank you for pointing this out. However, the 50% is not the performance criteria. But this is the prevalence of our outcome variable (academic performance) for sample size calculation, since we couldn’t find prevalence of poor academic performance in Hargeisa city, we have taken 50% as a default prevalence to calculate sample size as there was no previous study did in the area. 

3. Comments: [Describe what the acronym COR means in the section “Data processing analysis and analysis processing procedures”]. Response: Thank you, the COR is the Crude Odd Ratio, and we have, accordingly, described under that section. Also identified under “Abbreviation and acronym section”

4. Comments: [The results included data on the adolescents' lifestyle habits, information related to basic sanitation and hygiene, as well as data on other family members. However, this information was not presented in the methods. It is important to include a comprehensive description of these aspects in the methods, ensuring consistency in the presentation of methodological procedures and allowing a more complete understanding of the approach adopted]. Response: Agree. We have, accordingly, included other variables descriptions. Page number (6 & 7).

Results 

1. Comments: [In the method, the authors chose to categorize data related to food security and insecurity in a dichotomous way. However, the results show the presentation of both this categorization and the previous categories. I recommend that authors clearly establish which categories will be used consistently throughout the study: whether they will be Food security and Food insecurity, or whether a more detailed approach will be applied, classifying into Food security, Mild food insecurity, Moderate food insecurity and Severe food insecurity, providing greater clarity and uniformity in the analysis and interpretation of data.]. Response: Thank you for pointing this out. However, the first categorization of food security was for descriptive analysis purpose, but the further dichotomization to “food secure and food insecure” is the final category that we have used consistently throughout the study. Also, we have identified this throughout the manuscript. 

2. Comments: [Statistical analyzes were carried out to investigate the association between food insecurity, academic performance and the data presented in the categories “Adolescent illness and factors related to food” “Factors related to adolescent lifestyle” “Environmental sanitation and factors related to hygiene”? It is unclear how descriptive analysis of these data, as presented in the article, contributes to the study. I suggest considering excluding these data from the survey or, alternatively, providing statistical analyzes that demonstrate the relationship (or lack of relationship) between these factors and food insecurity and academic performance. This review may enhance the relevance and impact of the findings in the study]. Response: Thank you for this suggestion. However, we believe that most of the variables would be more appropriate because academic performance among adolescents can be affected by other factors. Thus, we implied with your suggestion throughout the manuscript. Although one limitation of our study is that it did not analyze the relationship between these factors and food insecurity, thus we have provided statistical analyses that shows the relationship between these actors and academic performance. 

3. Comments: [The excerpt below is part of the methods; I suggest describing our methods and extracting the results. “Bivariate analysis was performed to select candidate variables for multivariate regression analysis at P≤0.25. Thus, food security status, birth order, head of household, maternal education, paternal education, family wealth index, habitual breakfast intake, daily meal frequency, school absenteeism, home study duration, screen time, and practice of washing hands after using the bathroom were selected as candidate variables.]. Response: Thank you for this suggestion. We agree with this comment. We have incorporated your suggestions throughout the manuscript. 

Discussion

1. Comments: [Based on the results of the study, it is essential to enrich the discussion by incorporating some unaddressed points, which could add depth to the understanding of the topic. Suggest including in the discussion; - The importance and central role of the school in ensuring food security for students, taking into account that this is the environment where children eat most of their meals.- A more in-depth analysis of the results, including data on the family, the influence of maternal education and the dynamics of female-headed households in relation to food security.- A more detailed analysis of food consumption is recommended in the discussion, considering that this indicator plays a fundamental role in the assessment of food security. Note a limited approach to the discussion of food consumption, with only one paragraph dedicated to the debate about breakfast. Expanding the discussion to cover different aspects of food consumption would allow for a more comprehensive understanding of students' eating habits.]. Response: Thank you for pointing this out. We agree with this comment. We have incorporated your suggestions throughout the manuscript. 

2. Comments: [Describe the limitations as a discussion paragraph and add the strengths of the study]. Response: Thank you for pointing this out. We agree with this comment. We have included your suggestions throughout the manuscript. 

Additional clarifications: In addition to the above comments, all spelling and grammatical errors pointed out by the reviewers have been corrected. We look forward to hearing from you in due time regarding our submission and to respond to any further questions and comments you may have. 

Sincerely,

Sagal Mohamed Adam 

Sagalsama24@gmail.com

30.01.2024

---

## [Decision Letter · Decision Letter 1]

18 Apr 2024

Household food insecurity and its association with academic performance among primary school adolescents in Hargeisa City, Somaliland

PONE-D-23-31840R1

Dear Dr. Adam,

We’re pleased to inform you that your manuscript has been judged scientifically suitable for publication and will be formally accepted for publication once it meets all outstanding technical requirements.

Kind regards,

Larissa Loures Mendes, Ph.D.

Academic Editor

PLOS ONE

Additional Editor Comments (optional):

Reviewers' comments:

Reviewer's Responses to Questions

**Comments to the Author**

1. If the authors have adequately addressed your comments raised in a previous round of review and you feel that this manuscript is now acceptable for publication, you may indicate that here to bypass the “Comments to the Author” section, enter your conflict of interest statement in the “Confidential to Editor” section, and submit your "Accept" recommendation.

Reviewer #3: All comments have been addressed

Reviewer #4: All comments have been addressed

2. Is the manuscript technically sound, and do the data support the conclusions?

Reviewer #3: Yes

Reviewer #4: Yes

3. Has the statistical analysis been performed appropriately and rigorously? 

Reviewer #3: Yes

Reviewer #4: Yes

4. Have the authors made all data underlying the findings in their manuscript fully available?

Reviewer #3: Yes

Reviewer #4: Yes

5. Is the manuscript presented in an intelligible fashion and written in standard English?

Reviewer #3: Yes

Reviewer #4: Yes

6. Review Comments to the Author

Reviewer #3: (No Response)

Reviewer #4: (No Response)

7. PLOS authors have the option to publish the peer review history of their article (what does this mean?). If published, this will include your full peer review and any attached files.

Reviewer #3: No

Reviewer #4: **Yes: **Nayhanne Gomes Cordeiro

---

## [Editor Report · Acceptance letter]

31 May 2024

PONE-D-23-31840R1 

PLOS ONE

Dear Dr. Adam, 

I'm pleased to inform you that your manuscript has been deemed suitable for publication in PLOS ONE. Congratulations! Your manuscript is now being handed over to our production team.

Kind regards, 

on behalf of

Dr. Larissa Loures Mendes 

Academic Editor

PLOS ONE